# A Review of the Aquatic Environmental Transformations of Engineered Nanomaterials

**DOI:** 10.3390/nano13142098

**Published:** 2023-07-18

**Authors:** Daniel Mark Harrison, Sophie M. Briffa, Antonino Mazzonello, Eugenia Valsami-Jones

**Affiliations:** 1School of Geography, Earth and Environmental Science, University of Birmingham, Birmingham B15 2TT, UK; 2Department of Metallurgy and Materials Engineering, Faculty of Engineering, University of Malta, MSD 2080 Msida, Malta

**Keywords:** environmental transformations, engineered nanomaterial, chemical transformations, physical transformations, biological transformations, aquatic environments

## Abstract

Once released into the environment, engineered nanomaterials (ENMs) undergo complex interactions and transformations that determine their fate, exposure concentration, form, and likely impact on biota. Transformations are physical, chemical, or biological changes that occur to the ENM or the ENM coating. Over time, these transformations have an impact on their behaviour and properties. The interactions and transformations of ENMs in the environment depend on their pristine physical and chemical characteristics and the environmental or biological compartment into which they are released. The uniqueness of each ENM property or lifecycle results in a great deal of complexity. Even small changes may have a significant impact on their potential transformations. This review outlines the key influences and outcomes of ENM evolution pathways in aquatic environments and provides an assessment of potential environmental transformations, focusing on key chemical, physical, and biological processes. By obtaining a comprehensive understanding of the potential environmental transformations that nanomaterials can undergo, more realistic models of their probable environmental behaviour and potential impact can be developed. This will, in turn, be crucial in supporting regulatory bodies in their efforts to develop environmental policy in the field of nanotechnology.

## 1. Introduction

Engineered nanomaterials (ENMs) are intentionally produced nanomaterials (NMs) possessing unique physicochemical properties due to their small size [1]. More specifically, NMs are described as materials with at least one dimension in the nanoscale (1–100 nm). Their small size results in an increase in surface area to volume ratio, making NMs more chemically reactive than their bulk-scale counterparts [2]. Additionally, at the lower end of the nanoscale spectrum, NMs display quantum effects, whereby properties such as fluorescence, conductivity, magnetic permeability, and chemical reactivity deviate from standard bulk material behaviour and become a function of particle size [3]. These novel properties make ENMs desirable for several new applications, thus making them more frequently incorporated into consumer products such as cosmetics, paints, textiles, and electronics [4], as well as being utilised in industrial-scale processes and novel biomedical practices. Clearly, the rapid growth in ENM use translates into an increased presence of these materials in the environment [5]. Critically, in assessing ENMs’ environmental role, it is necessary to consider their complete life cycle. This encompasses all stages of potential transformations, from the moment of manufacture through consumer use and ultimate disposal [6]. The control and monitoring of ENMs is relatively straightforward during the manufacturing stage, and at this stage, the responsibility to provide safety data for pristine ENMs should be with the manufacturer [7]. However, product behaviour and transformation outcomes become far less predictable once nano-products fall into the consumer realm.

During their lifetime, ENMs may be disposed of in landfills and potentially find their way into soil, air, or aquatic environmental compartments. This can occur at different stages in their life cycle, such as during synthesis, product manufacturing, distribution, the use phase, or disposal. Due to their small size and reactive nature, ENMs may also be released from waste management facilities into surface waters rather than being retained within solid waste. It was estimated that 63.0–91.0% of over 260,000–309,000 metric tonnes of global ENM production in 2010 ended up in landfills, with the balance released into soils (8.0–28.0%), water bodies (0.4–7.0%), and the atmosphere (0.1–1.5%) [8]. Of those that are released into aquatic environments, titanium dioxide (TiO_2_), zinc oxide (ZnO), carbon nanotubes (CNTs), and silver (Ag) ENMs are thought to be the most commonly identified [6]. Consumer use of personal care products is a primary route for aquatic deposition, often via waste-water treatment plants (WWTPs), and this brings about the potential for a range of transformation reactions to occur.

Clearly, the increasing incorporation of ENMs into consumer products is matched by the growing need to improve our understanding of their behaviour and apply appropriate risk assessments. However, if we are unable to accurately monitor and predict ENM behaviour, then our ability to safely manage their risks is limited. This will reduce public confidence and affect the nanotechnology industry’s potential to thrive [9].

The prediction of ENM transformations is hindered by our limited understanding of the complex chemical, physical, and biological transformation processes ENMs can undergo. This is a consequence of their highly reactive nature and is especially pertinent in aquatic environments, which can be considered a major compartment for transformations to occur or a conduit for ENMs to transform during transport to other environmental compartments [10].

Transformations are dynamic reactions that are dependent not only on the intrinsic physicochemical properties of ENMs, such as size, composition, surface area, surface charge, and morphology, but also on external media drivers such as temperature, pH, ionic strength, and the presence of inorganic and organic aquatic species [5]. In recent years, several studies have attempted to incorporate more complex variables into mathematical models of ENM fate in aquatic systems to better understand these dynamic reactions. A study by Dale et al. [11] suggests that modelling has progressed from early developments of model flow analysis (MFA) to more complex ENM fate and transport (F&T) modelling. However, environmental conditions, including temperature, pH, ionic strength, etc., are still rarely included in models. There is a drive to produce more dynamic models that incorporate these variables in the hope that they will give a more accurate representation of real-life outcomes [11].

Another area of limitation for testing environmental transformations is the issue of kinetics. Many transformations are rapid processes, and yet others are kinetically sluggish, taking months or even years to occur. One study [2] used thermal transformations as a proxy for the ageing of a library of comparably sized laboratory synthesised ENMs with core compositions of copper oxide (CuO), zinc oxide (ZnO), and ceria (CeO_2_) and functional surface coatings of polyvinylpyrrolidone (PVP). These were then compared with a manufacturer-purchased, uncoated ceria dioxide. The thermal heating of the particles enabled the investigators to establish long-term environmental behaviour for these metal oxide ENMs over a shorter time period. The work showed that particle capping helped limit transformation, but even PVP capping of particles allowed a degree of physical and chemical ENM transformation to take place [2].

Amendments to regulations, including Registration, Evaluation, Authorization, and Restriction of Chemicals (REACH) Annexes, which aim to properly account for the distinct properties of ENMs, have recently been established in order to enhance the safety assessment of ENMs [12,13]. However, most safety test methods were originally designed for soluble bulk chemicals and are not well suited to ENMs. Thus, it is important that they be adapted or revised with the intent to properly assess suspensions of ENMs that do not readily dissolve or that may transform. In 2015, the Organisation for Economic Co-operation and Development’s (OECD) Working Party on Manufactured Nanomaterials (WPMN) listed environmental transformations as a knowledge gap, calling for the development of test guidelines and guidance documents specific to ENMs to support chemical regulatory systems. ENMs currently being used or newly entering the market must be adequately tested to ensure they will not result in a negative impact on the environment or environmental organisms. To understand their potential impacts and carry out a sufficient risk assessment, an understanding of how ENMs are released into the environment, how they transform, and how these changes affect their environmental fate and impact on organisms is needed. Currently, internationally standardised test methods to provide such crucial data to risk assessors are lacking.

This review outlines the key issue of ENM behaviour in aquatic environments and, through examples in the literature, provides an insight into the assessment of ENM environmental transformations, outlining key chemical, physical, and biologically mediated processes. Particular attention is given to aquatic media composition and its influence on solid-state and species-driven transformations. Though transformations are considered here as distinct mechanisms, multiple transformations typically occur simultaneously and at different rates. This is due not only to the high interdependence on ENM intrinsic properties, such as surface area and size, but also to external factors such as media composition, pH, temperature, and ionic strength acting in tandem [2,10]. In many instances, transformations are interconnected, whereby one process will facilitate another.

## 2. Chemical, Physical and Biological Transformations

### 2.1. Transformation Overview

Once released into the environment, ENMs are likely to undergo various transformations influencing their chemical, physical, and biological properties (Figure 1) [5]. Table 1 highlights the factors determining the outcome of transformations and the property or variable having the greatest control over the transformation. Furthermore, Table 2 gives examples of transformations noted for commonly used ENMs, specifically TiO_2_, CuO, CeO_2_, Ag, ZnO, and graphene, in aquatic environments. Table 3 gives the observations of the specific examples explored. All the transformations that ENMs undergo have the potential to alter their solid-state properties. This, in turn, can have significant impacts on environmental and toxicological behaviour. Only through a full understanding of their potential transformation pathways and changes can we anticipate the extent of the impact these ENMs may have.

### 2.2. Chemical Transformations

A chemical transformation is the conversion of a compound into one with a different structure, valency, or composition. There are numerous chemical transformations that can occur due to various internal and external factors. These can involve a change to the entire ENM; a change to the surface functionalisation, if present; or the formation of a core-shell material by surface reactions when only the top layer of the ENM core is modified [100].

Chemical transformations influence particle behaviour. Dominant chemical reactions in aquatic media are governed by redox (reduction/oxidation) reactions, dissolution, precipitation, and complexation, either as substitutions of parts/whole of the ENM or on its surface, the latter forming an environmental organic corona (Figure 2).

#### 2.2.1. Redox

Reduction-oxidation (redox) reactions are critical to ENM chemical transformations. These reactions result from an electron transfer between two different chemical moieties, as shown in Equations (1)–(3), resulting in new moieties that may be more or less reactive in that specific environment than their earlier counterparts [10,101].
Oxidation: A → A^x+^ + e^−^(1)
Reduction: B^x+^ + e^−^ → B(2)
Overall Reaction: A + B^x+^ → A^x+^ + B(3)

Redox reactions drive dissolution and complexation with other chemical constituents by altering the oxidation state of the ENM and thus the ENM’s reactivity. Oxidation and reduction reactions are highly influenced by the water chemistry of the environment that they find themselves in. Oxidising zones can be found in natural waters, surface environments, and aerated soils, while reduction areas can be found in groundwater, wastewater, and sediment. Redox cycling is very common in tidal zones due to their dynamic environmental conditions, which can result in a complex cycle of ENM transformations [5,100,101].

Redox reactions with ENMs are governed by the availability of dissolved oxygen (O_2_) or reducing agents (e.g., organic matter), but are also dependent on intrinsic physicochemical properties of ENMs such as size, surface charge, composition, and reactivity (Figure 3). In addition, redox chemical changes are temperature dependent [87]. For instance, Briffa et al. showed that cerium dioxide ENMs undergo valence state changes upon exposure to elevated temperatures [2]. They are also dependent on ionic strength, as observed by Tantra et al. when measuring the redox potential of six different sizes of nanomaterials, using an oxidation-reduction potential (ORP) electrode probe, in deionised water and various ecotoxicity media [87,88]. Furthermore, redox chemical reactions may be influenced by the presence of macromolecules and organic ligands from natural organic matter (NOM), as observed by Hoffmann et al. when studying cadmium sulphide nanoparticle growth and colloid stability [87,89].

Understanding the degradation and release of redox reaction products, such as dissolved species, can help elucidate solid-state transformations of the parent material as well as provide valuable toxicity data. For example, in oxidising aquatic environments, elemental silver (Ag^0^) ENMs are likely to undergo dissolution, facilitating the release of toxic Ag⁺ ions [10]. Such reactions are exploited in the textile industry for their biocidal properties. The dissolution of elemental silver is accompanied by the release of an electron, which can facilitate the production of damaging reactive oxygen species (ROS) [60]. This may be beneficial in commercial products such as cleaners and detergents, or bio-textiles like antibacterial socks [102]. Furthermore, biogenic Ag nanoparticles have been successfully used to induce cytotoxicity in human-derived cancer cells through oxidative stress mechanisms [103]. However, increased incorporation of these materials in consumer products and in the biomedical industry will lead to increased incidental releases into the environment [5]. The uptake and internalisation of ENMs within cells has the potential to cause detrimental cellular effects and could pose a risk to humans and aquatic species [104]. It has been shown that internalisation of ENMs causes ROS production, the extent of which may also be ligand-specific. For example, it has been shown that both positively and negatively charged ligands bound to gold ENMs are able to induce ROS production. There is a much higher percentage of ROS produced by *Daphnia magna* (*D. magna*) enterocytes that are exposed to positively charged cetrimonium bromide (CTAB)-coated gold ENMs compared to negatively charged citrate-coated ENMs [105]. Oxidative stress is a commonly used marker for NM-induced cellular stress. ENMs may be internalised by organisms via endocytic or passive processes, allowing the delivery of toxic dissolution products to cellular constituents, the so-called “Trojan Horse” effect [1].

Redox reactions are dependent on the media composition and conditions. In addition to oxygen dependence, these reactions may also be affected by pH if the ENM transformation involves protons or hydroxide ions. Dissolution of Ag, for example, is significantly enhanced as pH is reduced. This is because of increased Ag solubility in more acidic environments. It has been shown that for Ag in naturally oxidised aquatic environments, an increase in both the potential of electrochemical reactions and pH typically leads to the precipitation of AgCl complexes. This contrasts with sulfidation, which occurs in natural waters isolated from the air or in anoxic environments such as WWTPs. In the intermediary zone between oxidised and reduced water systems, Ag typically exists as solid elemental Ag^0^, though systems rarely stay in equilibrium. Additionally, this will depend on the concentration of other system variables, such as chloride and sulfide. An increase in concentration will have a proportional effect on the rate of transformation, while the presence of organic ligands, including NOM, may slow down these processes [60]. Transformations to Ag_2_S species are likely to be limited to terrestrial environments, as seawater rarely reaches low enough potential values due to its aerobic nature.

Overall, it can be noted that redox reactions are very common and likely to occur in the environment. However, by understanding the potential life-cycle pathways of ENMs that may undergo redox reactions, including whether these ENMs will be found in oxidizing or reducing environments, it may be possible to implement a safety-by-design approach to take these redox transformations into consideration. This means that scientists should aim to design safer ENMs through aspects such as core-shell modification or capped particles, whereby the surface is more resistant and resilient. This information can also help inform governing bodies for regulatory purposes.

#### 2.2.2. Dissolution

Dissolution, which is the formation of a solution by dissolving a solute in a solvent, is inherently difficult to assess. Variables such as experimental set-up, contamination, heterogeneity of samples, and structural defects can make the quantification of dissolution challenging [87]. In principle, measuring dissolution involves analysing the concentration of a dissolved ionic species in a specific medium over a given period of time [10]. Filtration and ultracentrifugation are commonly employed for the purposes of separation of the dissolved and non-dissolved phases [7]. However, many of the techniques already established require some adaptations to account for nanoscale properties.

Their small size means that ENMs will simply pass through many of the filters designed for bulk-material separation. One of the major difficulties is the separation of dissolved ionic species from particulate matter. Membrane filtration through either ultrafiltration or dialysis membranes relies on the passage of material through filters. These are most commonly 0.2 or 0.45 µm filters, with the key difference being that the former is forced through the membrane with pressure, whereas the latter separates based on concentration gradients [87]. In addition, although there will likely be an element of ENM retention on filter papers or membranes, the nominal pore size may allow individual nanoparticles through the filter pores [87]. An investigation considering the effect of pH, particle size, and crystal form on dissolution found that membrane dialysis gave comparable results to the syringe filter method but took too long to complete, thus compromising the experimental design [106]. Therefore, it is important to consider not only accuracy, precision, and experimental reproducibility but also practicality and appropriateness within time constraints for a given experiment.

Coupling these processes with analytical mass spectrometry methods, such as inductively coupled optical emissions spectrometry (ICP-OES) and inductively coupled plasma mass spectrometry (ICP-MS), may prove useful for determining dissolved ionic species content. While in both techniques an argon plasma is used to ionise the elements in a sample, ICP-OES uses the emitted radiation from the plasma to analyse the excited atoms, while ICP-MS uses MS to measure the ions’ mass [107]. One main difference between the techniques is that ICP-MS has a lower detection limit extending to parts per trillion, while the lower limit for ICP-OES is parts per billion. However, ICP-OES has a higher tolerance to total dissolved solids (up to 30.0%) when compared to ICP-MS (about 0.2%). For determining dissolved ionic species content, complete separation of particulate matter and dissolved ionic content needs to be guaranteed, as, except for single-particle ICP-MS methods, mass spectrometry instruments identify ions based on mass and are unable to distinguish between dissolved species and particulate matter.

Determining the dissolution rate of ENMs is important to understand changes related to their bioavailability, redox activity, fate, and toxicity. Dissolution is often used as a measure of a nanomaterial’s biodurability and, in turn, toxicity [108]. Lee et al., for instance, found that dissolution influenced the toxicity of citrate-stabilised silver ENMs and polyethylene glycol-coated silver ENMs to zebrafish embryos in two different ionic environments [90]. Unfortunately, to date, many studies on ENM dissolution have not determined dissolution rates and rate constants [108]. One of the few works to have done so is that of Yadav et al., who found that dissolution is affected by the water’s characteristics [43]. The authors showed that CuO NPs were more soluble in deionised water (0.054 mg/L) than in natural pond water (0.035 mg/L); however, the dissolution rate was faster in pond water (0.049 h^−1^) compared to deionised water (0.034 h^−1^) [43]. This highlights how dependent transformations are on the environment. Supporting this dependency on the environment, Stetten et al. [29] found that under oxic conditions, ZnO NPs were dissolved within a few hours; however, under anoxic conditions, dissolution was much slower. In the study of Yadav et al., the dependence of transformations on the chemistry of the NP in question is also highlighted. Natural pond water consists of NOM, minerals, and other contaminants that could affect the dissolution and dissolution kinetics. NOM can produce a protective coating on NPs that mitigates their dissolution. Decreasing the pH and increasing the ionic strength of the medium enhances the dissolution of ENMs. Indeed, Mbanga et al. showed that the dissolution rate of Ag NPs increased at high ionic strength and low pH [44].

The dissolution rate of ENMs in environmental media is a regulatory information requirement within REACH and of importance in other chemical regulations worldwide [109]. Some data and methods are available on the dissolution kinetics of metal and metal oxide ENMs and can be used for the development of an ENM-specific dissolution test guideline (TG). However, current methods for the determination of dissolution rates were primarily designed for bulk dissolvable chemicals such as OECD TG 105 [110]. Few methods that take the nanodimensions into consideration actually exist (e.g., ISO 19057). Extensive consideration of available methods has concluded that, when implemented for ENM dissolution testing, they all have associated advantages and disadvantages. TG 105 describes two methods (a static batch test and a dynamic test) to determine solubility, which can be adapted for ENMs [110]. The recent (2021) OECD guidance document 318 which focuses on the dispersion stability of nanomaterials, is the first step towards reliably assessing their behaviour in media [111]. These tools would help support governments and regulators in implementing effective risk assessment policies for ENMs.

In recent years, there have been dedicated research efforts to provide approaches and data to support the development of the TG. There are some comprehensive studies that have been carried out in relation to dissolution, such as the work of Keller et al. [112] and Koltermann-Jülly et al. [113]. Overall, there are multiple ongoing activities relating to dissolution work being conducted as part of EU-funded research projects and at the OECD. The multifold dimensions and complexity of developing such guidelines for such complex and transformational materials need to be recognised.

#### 2.2.3. Structural Transformation

##### Sulfidation

Redox reactions can initiate species-driven chemical changes such as sulfidation. This is an important consideration, as although in this review transformations are discussed separately, in reality, multiple transformations can happen concurrently and/or sequentially. Having knowledge about ENM physicochemical properties and media parameters, such as dissolved O_2_, pH, ionic strength, and solubility constants, enables predictions to be made using geochemical speciation models and phase diagrams [10].

One study considered the effect of sulfidation on Ag ENMs and its impact on dissolution [71]. The authors used deionised water and various biological media complemented with artificial seawater. They found that in all cases, partly s Ag ENMs released fewer toxic Ag⁺ ions than pristine Ag ENMs over 48- and 120-h intervals and that this reduced toxicity to aquatic organisms. Levard et al. [71] also found that toxicity was significantly reduced in the higher ionic strength solutions compared to those of lower ionic strength. It was reported that this was independent of sulfidation and likely to be due to chloride complexation with dissolved species to form AgCl species [71]. Zhang et al. [91] found that increasing the presence of NOM actually suppressed the sulfidation of Ag nanowires in the aquatic environment. The authors [91] explained that the Ag nanowires could coordinate and be complex with the N and S-containing ligands in the NOM, forming strong-bonded coatings that would decrease the dissolution of the Ag nanowires and thus their interaction with sulfides. Furthermore, Zhang et al. [91] found that the zeta potential of Ag nanowires in the presence of NOM was more negative than that of pure Ag nanowires. Hence, the electrostatic repulsion between the negatively charged Ag nanowires and sulphide ions increased in the presence of NOM, with the consequence that the sulfidation rate was reduced. Meanwhile they [91] reported that the presence of divalent cations, including Mg^2^⁺ and Ca^2^⁺ in solution accelerated sulfidation rates when compared to monovalent ions such as Na⁺ and K⁺. Other work [72] considered the effect of sulfidation on PVP-coated Ag ENMs, and, again, sulfidation was found to reduce dissolution and limit toxicity. This study also showed that the PVP coating failed to protect the Ag particles from corrosion and, consequently, elemental silver (Ag^0^) was oxidised upon reacting with sulphur (HS^−^) and then re-precipitated as Ag_2_S nano-bridges between adjacent particles, forming aggregates with a lower dissolution potential.

Prasher et al. concluded that smaller AgNPs could result in an enhanced sulfidation rate owing to the reaction rate dependency on the specific surface area of the NP and found that the increased HS^−^/Ag ratio also significantly enhanced sulfidation [75]. The presence of NOM was also found to influence the sulfidation of AgNPs. In contrast to the findings of Zhang et al. [64], Zhang et al. observed that the presence of HA promoted the sulfidation of AgNPs by replacing the surface coating of NPs, thus increasing the available surface area [68]. Similarly, Thalmann et al. [70] found that the sulfidation rate increased with increasing NOM concentrations. The discrepancy with the observations made by Zhang et al. in [64] could be explained by the different morphologies of the Ag nanomaterials. For AgNPs, their stability was promoted and their aggregation reduced by NOM [68]. As a result, the surface area of AgNPs increased, such that more AgNPs were able to react with sulphide ions [68]. In the case of Zhang et al. [64], the Ag nanowires were already well dispersed in the absence of NOM. Thus, their surface area was unaffected by NOM. Rather, the absorption of NOM on the Ag nanowires reduced sulphide ions’ access to react with Ag. Sulfidation is usually accompanied by substantial aggregation and sedimentation and a lower dissolution rate, inevitably affecting the bioavailability and fate of NPs. Indeed, Cao et al. reported that sulfidation mitigated the toxicity of AgNPs in constructed wetlands [76].

ZnO ENMs also have the potential to release toxic Zn^2^⁺ via dissolution [26]. It has been shown that under ambient temperature conditions and in the presence of sufficient concentrations of sulfide, the ZnO ENMs would completely transform to ZnS_(s)_ within 5 days. The authors suggested that this is a highly likely transformation in the anoxic river and lake sediment environments, where typical sulphide concentrations range from 1 to 100 µgL^−1^ [26]. In wastewater treatment facilities, sulphide concentrations are expected to rise from a few µgL^−1^ to 10 mgL^−1^, further increasing the likelihood of occurrence. This is important for Zn ENMs, as many of the consumer products that contain ZnO ENMs are likely to pass through water-treatment facilities prior to being discharged into various aquatic environments such as rivers, lakes, seawater, and WWTPs. Critically, stabilisation of ZnO ENMs through sulfidation can reduce toxicity as it reduces dissolution and ion release [26]. Indeed, Lee et al. showed that sulfidation of ZnO NPs reduced embryonic zebrafish toxicity, which was attributed to the hindrance of zinc release by the sulphates that probably enclosed the NPs [28]. However, this also increases environmental persistence.

##### Phosphatisation

Phosphatisation has a similar mechanism to sulfidation but is instead driven by the phosphate constituents within the media. Phosphatisation of ZnO ENMs has been shown to be pH-dependent and more likely in acidic environments than in alkaline environments [92]. Xu et al. [92] demonstrated that phosphatisation at the surface of ZnO ENMs greatly enhanced their ability to remove harmful lead ions in polluted water. This highlights that artificial or natural phosphate capping of ENMs can be beneficial for the remediation of trace metal-polluted waters, such as those in acid mine drainage (AMD). Similar to sulfidation, the phosphatisation of ENMs can mitigate toxicity in microorganisms. For instance, Lee et al. reported a decrease in toxicity in embryonic zebrafish when inducing ZnO NP’s transformations with phosphor [28].

Lv et al. found that the addition of a low concentration of phosphate to ZnO ENMs resulted in an altered morphology when observed with TEM [31]. The ZnO ENMs were transformed from uniform spheres (30–50 nm) to amorphous and aggregated crystalline phases. This was further investigated with X-ray absorption near edge structure (XANES) showing the structural transformation from ZnO to hopeite (Zn_3_(PO_4_)_2_·4H_2_O) [31]. Phosphate transformation products for ZnO ENMs are larger than their pristine counterparts, and thus, surface reactivity is decreased, leading to reduced dissolution and muted toxic potential, as well as the potential to scavenge harmful trace metals through adsorption onto porous phosphate surfaces [31].

Phosphatisation is also highly relevant to other metal-oxide ENMs. Zhang et al. [55], Briffa et al. [54], and Romer et al. [53] have all shown that physical and chemical changes occur when cerium dioxide (CeO_2_) nanoparticles are exposed to phosphate-rich conditions at pH 5.5 [53,54,55]. Figure 4 shows an example of the change in shape and morphology to sea-urchin-like structures as zirconium-doped cerium dioxide ENMs are exposed to a phosphate solution at pH 5.5 [54].

Hartmann reported that CeO_2_ was found to aggregate in aquatic environments, typically depositing within the soil sediment through adsorption. However, increased concentrations of phosphate will encourage the desorption of CeO_2_ ENMs from natural colloids [10]. thus limiting environmental CeO_2_ persistence and reducing the toxicity risk to some organisms, particularly filter feeders. Work by Zhang et al. demonstrated that phosphate was capable of immobilising CeO_2_ through phosphate complexation in plant roots [93]. The authors found that this inhibited further translocation and toxicity from CeO_2_ and that plant biomass was reported to be higher for samples treated with phosphate.

##### Carbonation

Carbonate (CO_3_^2−^) concentration in different environmental media plays a significant role in driving carbonate complexation and precipitation [10]. Carbonation also affects particle stability. For example, inorganic silver carbonate (Ag_2_CO_3_) coatings have been applied as capping agents to Ag ENMs to stabilise them against aggregation. In addition, these coatings can be acquired naturally following release into aquatic environments [60]. Natural precipitation of Ag_2_CO_3_ at particle surfaces is only likely in alkaline environments, as metal carbonates are unstable at pH below 7 [60]. Carbonation is, therefore, unlikely to occur in many of the natural environments in which Ag ENMs may be released [60]. Piccapietra et al. demonstrated that at an alkaline pH, negatively charged CO_3_^2−^ surface capping could inhibit aggregation [78]. However, they found that when increased concentrations of divalent Ca^2^⁺ (more than 2 mM) were introduced to the media and pH was reduced below 5, CO_3_^2−^ species were neutralised, moving the zero-point of charge close to zero and resulting in agglomeration [78].

#### 2.2.4. Surface Corona Reactions

Upon being released into the environment, ENMs have the potential to interact with other substances such as macromolecules, other NOMs such as humic substances [10], and biomolecules such as proteins or carbohydrates that are released by organisms as part of natural processes. These processes are typically characterised by adsorption and desorption, often resulting in surface coatings analogous to protein or environmental coronas [5].

Proteins also play a crucial role in the transformation of ENMs released into the environment, as they will react according to the surface charge of the proteins. Zhang et al. recently demonstrated that positively charged proteins enhanced the dissolution and sulfidation of AgNPs when compared to negatively charged proteins [68].

These interactions can lead to scenarios where ENMs either adsorb a substance such as humic acid (HA) onto their surfaces or become adsorbed onto other environmental substances, typically larger colloids [10]. Clearly, these processes facilitate transformation, altering the surface properties of the ENMs and having the potential to impact mobility, behaviour, and eventually environmental fate.

Yang et al. investigated humic acid adsorption behaviour for ZnO, TiO_2_, SiO_2_, and Al_2_O_3_. HA adsorption occurred for all ENMs except for SiO_2_, and the authors propose that electrostatic charge differences were responsible for the inability of SiO_2_ to adsorb HA due to paired repulsive negative charges [14]. The authors reported that the highest rates of adsorption were observed for TiO_2_ and ƴ-Al_2_O_3_, with ZnO and α-Al_2_O_3_ to a lesser extent. Adsorption of HA was dependent on pH and decreased as the solution became more basic [14]. Again, this is likely to be due to the shift in the zero point of charge. In another study, Kansara et al. showed that HA and clay stabilised TiO_2_ NPs compared to bare TiO_2_ NPs and TiO_2_ NPs with clay [18]. The latter were toxic to zebrafish embryo development, while TiO_2_ NPs in the presence of HA displayed a protective effect [18]. The authors attributed the beneficial effect of HA to a combination of hetero-aggregation and adsorption of HA, which decreased the availability of the TiO_2_ NPs [18].

Work by Diegoli et al. considered the effect of using the Suwannee River Humic Acid Standard on acrylate and citrate-capped gold nanoparticles [94]. They found that in low-ionic-strength solutions, Suwannee River Humic Acid provided an additional coating to capped gold ENMs, thereby providing additional resistance from pH-induced aggregation. However, it was found that humic acid did not prevent aggregation in the presence of high-ionic-strength solutions, which would be comparable to hard-water environments [94]. Clearly, these processes are technically challenging to quantify in natural environments due to the outweighing likelihood of heteroaggregation and variable external media parameters. These are all factors that need to be kept in mind when developing risk and testing guidelines.

Adsorption of molecules, including biomolecules, to ENMs’ surfaces alters the identity, stability, and toxicity of the ENM towards organisms. NOM and biomolecules may replace surfactants or ligands that originally provided charge or coating to the ENM surface in order to ensure dispersion [114]. The replacement of these ligands by NOM or biomolecules may cause stabilisation or destabilisation, leading to ENM transformation and possibly agglomeration. The presence of this environmental corona also transforms the ENM surface, as newly acquired biomolecules may result in a change in ENM shape, thereby also altering toxicity. For instance, Qin et al. showed a reduction in the toxicity of ZnO NPs in the presence of HA [35]. While the exposure of zebrafish embryos to ZnO NPs displayed toxicity effects, the presence of HA resulted in an increase in survival rate as a result of the reduction of adhesion of ZnO NPs on the embryonic chorionic surface [35]. Shape and morphology have been shown to have an impact on solubility as specific surface areas are influenced. Smaller ENMs are more likely to be energetically unstable and prone to dissolve [115] due to a larger available surface area and proportionally more atoms on the surface.

#### 2.2.5. Photochemical Transformation

Light may cause excitation of ENMs, free radical formation, and/or changes to an ENM’s surface/coating. Thus, changes can be due to incident light wavelength, the ability of light to penetrate the outer layers of the material, and photosensitivity or photo-degradation potential.

Photochemical reactions can have significant effects in the case of several metal oxide ENMs, in particular TiO_2_, and to a lesser extent Ag, CuO, and CeO_2_ [10]. Natural UV radiation must be accounted for when considering aquatic environments such as rivers and seawater. In addition, a growing number of wastewater treatment plants utilise artificially induced UV for water disinfection treatments [116].

Photochemical transformations are a form of chemical reaction in which materials absorb light [10]. The main transformation that occurs is the degradation of natural surfaces or artificial surface coatings present on ENMs. There are two principal types of photochemical degradation: photolysis and photocatalysis [10]. Photolysis is characterised by the absorbance of photons, which excites electrons to higher energy bands. As excited electrons are inherently unstable, they will emit radiation as they fall back to their ground state. It is at this point that the release of radicals such as ROS takes place. The release of ROS has the subsequent effect of damaging molecular bonds and, hence, causing the breakdown of surface structures on ENMs [10]. Photocatalysis on the other hand occurs in the presence of a catalyst, which can speed up chemical reactions [10]. Once again, the production of ROS and damage to surface structures are associated with this mechanism.

Nano-TiO_2_ is well known for its photoactivity and ability to adsorb UV radiation [20]. This is one of the main reasons it is used in cosmetics, particularly in sunscreen products. However, due to the potential that TiO_2_ has to generate ROS and oxidative tissue damage, much concern has been raised about the implications for toxicity and environmental impact [22]. An investigation by Sun et al. found that UV irradiation of TiO_2_ in the environment significantly increased aggregation [21]. They found that 50 h of UV irradiation accelerated the aggregation rate by 27 times, resulting in hydrodynamic aggregate diameters greater than 620 nm. Clearly, this transforms nano-TiO_2_ to result in a size that is outside the regulatory definitions for ENMs. Furthermore, the authors found the growth of hydroxyl surface groups as identified with attenuated total reflectance Fourier transform infrared spectroscopy (ATR-FTIR) [21]. ATR-FTIR provides information related to the presence of specific functional groups. Shifts in the frequency of absorption bands and changes in relative band intensities indicate changes in the chemical structure [117]. Thus, ATR-FTIR spectroscopy can be used to determine the resultant surface chemistry, especially following induced chemical or physical modifications [117]. Sun et al. [21] found that by increasing the UV irradiation time, successive growth of three IR bands at 3630, 3670, and 3730 cm^−1^ was observed, which correspond to the presence of hydroxyl groups. These changes to surface charge are expected to be the causal factor for rapid aggregation in this study via electrostatic de-stabilisation.

Studies on graphene oxide (GO) reveal that GO can undergo photochemical reactions when exposed to sunlight in natural environments [84]. Simulated sunlight can rapidly reduce the GO, producing by-products in the form of CO_2_ and low-molecular-weight species. In addition to degradation, photochemical reactions can result in the synthesis of ENMs from precursor materials in aquatic environments. Hou et al. suggested that NOM-facilitated photo-reduction of ionic Ag in river water could precipitate nano-sized Ag particulates of different sizes and morphologies [84]. They proposed a mechanism for Ag ion reduction when adsorbed onto NOM [84]. This is especially anticipated in river waters fed by municipal waste and industrial effluent. Importantly, photochemical reactions can occur over variable timescales. The study by Hou et al. found that the initial photoreaction and CO_2_ production from GO photoreduction were rapid and independent of dissolved O_2_ [84]. However, dissolved O_2_ was critical in driving photo-reduction over longer time periods (>10 h) when compared to nitrogen-saturated environmental conditions [84].

Another study by Hwang and Li considered the photochemical transformation of nano-C-60 under environmentally relevant conditions [95]. They found that the particles underwent surface oxidation and hydroxylation in the presence of dissolved O_2_ when investigated with X-ray photoelectron spectroscopy (XPS) and ATR-FTIR. However, no core change or dissolution occurred during the 21-day study. This is likely due to most photochemical reaction products being water-insoluble and prone to remaining on the surface of the particle [95]. They also found that photochemical transformations in the absence of dissolved O_2_ were negligible [95].

### 2.3. Physical Transformations

Physical transformations are essentially aggregation and agglomeration. The interactions between particles in solution are broadly governed by the DLVO theory (Deryaguin, Landau, Verwey, and Overbeek), which states that the interaction energy between particles in solution is the sum of the repulsive electrostatic forces and attractive forces [10]. Aggregation and agglomeration are particularly dominant physical transformations that can occur at any time throughout an ENM life cycle and can result in the conformation of a particle community [6]. Agglomeration and aggregation differ in that agglomeration involves the reversible conformation of particle groups under the weak attraction of Van der Waals forces, whereas aggregation is characterised by irreversible clustering due to strong chemical or electrostatic bonds (Figure 5). Aggregation and agglomeration are strongly dependent on the intrinsic physicochemical properties of the ENMs, such as size, surface charge, and capping agent, though they are also strongly influenced by the aquatic solution chemistry [118]. Once again indicating the importance of having a thorough understanding of ENM chemical and physical properties along with potential environmental pathways makes for predicting and understanding potential transformations, behaviour, and ultimately toxicity.

Since in environmental systems the concentrations at which ENM dispersions will be found are likely to be dilute, homoagglomeration and/or homoaggregation, which occur between the same type of NM, are unlikely, and heteroagglomeration and/or heteroaggregation, which occur between different types of NMs, will tend to dominate. This may result in changes in transport, dissolution, reactivity, and bio-uptake, as have been recorded for NPs including silver [101,119] and zinc oxide [65,101,120].

Aggregation typically leads to an alteration in the fractal geometry of ENMs with the potential to increase size and reduce the surface area to volume ratio, as well as increasing environmental persistence through reduced rates of dissolution [5]. Importantly, a subsequent decrease in reactivity may reduce environmental toxicity. Surface area and toxicity are particularly important, as any reduction in available surface area will reduce the available surface for ROS generation [121].

In most cases, ENM manufacturing involves the addition of stabilising functional surface coatings, specifically to prevent or minimise aggregation. However, the process of ageing has been shown to degrade the ENM surface coating. Kirschling et al. demonstrated that surface coatings such as polyethylene oxide can be lost through biodegradation, which ultimately results in aggregation [96]. Both TiO_2_ and ZnO are liable to photochemical oxidation, which can degrade particle coatings and induce aggregation [10].

Briffa et al. found that using temperature as a proxy for ageing led to enhanced degradation of PVP coatings for various metal oxide NPs, including CeO_2_, CuO, and ZnO [2]. Despite this, even at a high temperature (~80 °C), capped ENMs still provided greater resistance to the effects of aggregation than their uncapped counterparts. The pH of the medium has also been shown to influence aggregation since it controls the oxidative dissolution and surface charge of ENMs. For instance, Fernando and Zhou showed different behaviour for Ag NPs at low and high pH. Lower aggregation and higher particle stability were reported with increasing pH [80].

Qiu et al. observed that the aggregation and sedimentation of CuO NPs in soil solutions were influenced by the size of the NPs as well as the soil properties [48]. The authors found that larger-sized (80 nm) CuO NPs tended to settle without aggregation or via hetero/homo-aggregates, while smaller-sized (50 nm) CuO NPs formed hetero-aggregates with natural colloids in the soil, and the rate of sedimentation was accelerated [48]. The presence of dissolved organic matter increased the residual concentration of CuO NPs in soil solutions and thus their mobility [48].

### 2.4. Biological Transformations

Biological transformations in environmental media are highly relevant to solid-state transformations. These transformations share many similarities with the mechanisms mentioned for “interactions with organic media”, in which adsorption and desorption of macromolecules and organic ligands are key [10]. However, interaction with living organisms is also necessary to fulfil the definition of a biological transformation. These are changes that occur in the ENM and are related to living organisms or tissues (intercellular or extracellular) and environmental media [5,101].

Biological interactions, in the same way as physical interactions, can affect the surface charge, aggregation state, and reactivity. These could occur in the core or the coating and possibly change the NMs’ transport, bioavailability, and toxicity [5] by affecting surface charge, aggregation, and reactivity [100]. Biological transformations, such as bio oxidation, biodegradation, and interaction with macromolecules and organic ligands, including NOM and humic substances, have the potential to influence bioavailability, environmental reactivity, eco-persistence, and toxicity and, therefore, cannot be ignored [5].

Biological transformations fall into two broad categories: biodegradation and biomodification (Figure 6). Biodegradation occurs when microorganisms cause the breakdown or decomposition of materials such as graphene oxide [10]. In contrast, biomodifications are biologically mediated processes that are facilitated by organisms’ uptake or as an indirect consequence of interactions with living matter [10].

#### 2.4.1. Biodegradation

Biodegradation occurs when microorganisms, such as bacteria and fungi, break down compounds. Biodegradation is well documented for several carbon-based nanoproducts, including fullerenes (C-60) as well as single-wall and multiwall carbon nanotubes (SWCNT/MWCNT). One study using the OECD Ready Biodegradability Test (OECD TG 301) showed that C-60 (fullerenes) were not susceptible to biodegradation and that this was most likely due to its cage structure [10]. However, the sheet-like arrangement of carbon atoms in graphene may result in it becoming more susceptible to biodegradation and changing into more water-dispersible forms.

In contrast, other studies that were not conducted according to standardised OECD test guidelines report positive results on biodegradation. For example, single-walled carbon nanotubes were incubated with horseradish peroxidase and H_2_O_2_ via enzyme catalysis [97]. Clearly, this shows that the type of test method employed can be critical to the outcome of the transformation and can potentially lead to classification errors. Therefore, standardised testing protocols are vital to obtaining data to guide regulatory bodies. Some recent advances have been made in determining graphitic material degradation. Chen, Qin, and Zeng found that using naphthalene-degrading bacteria and associated enzymes resulted in the successful degradation of graphene oxide, graphite, and reduced graphene oxide materials [86]. Considering the increased use of carbon-based ENMs, this is encouraging as it offers a biological tool that can potentially be used to remediate waters polluted with carbon-based materials.

#### 2.4.2. Biomodification

Biomodification is a transformation process that occurs following uptake by an organism or due to an indirect biotic process. These processes may occur due to ingestion by organisms or through uptake into plants through roots [10,55]. A subset of examples from the numerous biomodification studies in the literature are discussed below.

Translocation and transformations in plants have been shown to be highly dependent on plant species and environmental concentrations of phosphate [93]. For example, a study examining CeO_2_ translocation and transformation found that where phosphate was removed from the media composition, plant translocation of Ce^3^+ from root to stem was enhanced, particularly for wheat and corn species. This is likely due to the immobilising effects of the phosphate, which was removed [93]. When the nutrient solution contained phosphate, pristine octahedral CeO_2_ formed rod-shaped complexes on root surfaces during immobilisation. In the absence of phosphate from the nutrient media, nano-CeO_2_ flocculated as aggregates in the root vacuoles [93]. This strongly suggests the interconnecting link that phosphatisation and biomodification have on nano-CeO_2_, especially with respect to facilitating chemical species changes and physical aggregation. In addition, this is of critical importance for aquatic environments, which receive high amounts of discarded phosphate-containing fertilizers.

ENM uptake and transformations both have an effect on the organism. One study found that *D. magna* were able to ingest lipid-coated (lysophophatidylcholine) nanotubes, using the lipid coating as a food source [98]. This resulted in the degradation of the lipid coating, which enabled the CNT to aggregate, aiding in their destabilisation.

Another study considered the acquisition of an eco-corona by polystyrene NMs [99]. *D. magna* neonates were exposed to a conditioning medium, allowing *D. magna* to secrete organism-specific proteins into the solution. The neonates were then removed from the conditioning media, and the polystyrene NPs were added. It was found that the polystyrene NPs quickly acquired specific *D. magna* macromolecular coronas on their surfaces, which induced aggregation and heightened uptake and gut retention [99]. In a separate study, Zhang et al. found that NPs coated with extracellular polymeric substances (EPS), organic polymers produced by microorganisms, formed EPS-NP coronas, which increased the accumulation of heavy metals and the biotoxicity of NPs [17]. The authors revealed that EPS coronas on TiO_2_ NPs and CeO_2_ NPs adsorbed more heavy metals (Cd^2+^, Pb^2+^, Cu^2+^, Ni^2+^, and Ag^+^) compared to NPs without EPS coronas [17]. Whilst such results showed the ability of EPS-NPs to absorb and remove heavy metals by forming metal complexes, they can also increase the toxicity of NPs by accumulating highly toxic heavy metals in marine environments [17].

Another study considered the effect that aggregation had on uptake by suspension-feeding bivalves. The study found that 100-nm ENMs were much more readily ingested as aggregates than the same 100-nm fully dispersed ENMs [122]. This also led to a longer gut retention time, which enhanced the likelihood of organismal toxicity. Clearly, the main ways in which organisms are likely to change the solid-state characteristics of ENMs are through surface coating enhancement or removal or through the acquisition of a biological or ecological coating. These transformation processes typically occur simultaneously with other transformation processes and often instigate secondary processes, including aggregation or morphological changes.

The main characteristics of the transformations reviewed have been summarised in Table 4.

## 3. Detection and Quantification of Transformations

Apart from understanding the potential transformations ENMs may undergo, it is important to be able to detect and quantify these transformations. The environmental transformations of ENMs are similar to those of bulk materials, yet they are usually enhanced due to nano-specific properties [6]. Therefore, various techniques are available to assess ENM transformations, many of which have been adapted from solid-state characterisation for bulk-materials including microscopy, spectroscopy, chromatography, and filtration [10]. Table 5 associates the detection method with specific ENM categories and highlights the advantages and disadvantages of the methods. This table is not meant to be comprehensive but rather give a general overview, and interested readers could refer to the work of Surangi et al. for a more comprehensive overview.

Electron microscopy, including transmission electron microscopy, scanning electron microscopy, and scanning tunnelling electron microscopy, allows for direct comparisons to be made between pristine and aged (transformed) ENMs. Microscopy techniques are highly valuable and allow the size, morphology, and surface conditions of ENMs to be assessed [123]. While extremely useful for qualitative and semi-quantitative analysis of transformational change, microscopy has some limitations. For instance, acquired images give no true indication of the sample dispersion state prior to sample preparation and drying, and the effect of smaller particles overlaying larger particles can distort measurements [124]. Moreover, accurate determination of primary size and size distribution requires that a minimum number of particles be measured to achieve statistical relevance. Microscopy is limited by numerous factors, including the 2D representation of a 3D material, operator bias, lower size resolution, and non-uniformity of environmental samples [7]. Lower detection limits can seriously impede the characterisation of ENMs, as a significant particle number concentration could be effectively “invisible” to the instrument [7]. Consequently, this could result in an inappropriate classification of ENMs and in turn an ineffective risk assessment. In addition, these microscopes can be coupled with ancillary instruments such as energy dispersive X-ray spectroscopy (EDS) and electron energy loss spectrometry (EELS), enabling data acquisition for the surface distribution of elements as well as oxygen-state determination. Microscopy is critical for morphology, size, and size-distribution data, which are vital for ENM classification under REACH.

Spectroscopic analytical instruments routinely used in ENM characterisation can offer suitable assessments for ENM transformation. For instance, dynamic light scattering (DLS) provides information about the hydrodynamic size (Hd) and zeta (ζ)-potential and allows the user to record any change between pristine and aged particles. However, the key limitation here is that DLS records only hydrodynamic diameter, making it impossible to distinguish between particles in isolation and aggregated clusters, and, in addition, it is only suitable for spherical-shaped particles [124]. Further to this, DLS cannot distinguish between particles of different compositions within mixtures or clearly distinguish a mixture of sizes [124]. In contrast, nanoparticle tracking analysis (NTA) allows visual tracking of particles through time hence aggregates can be clearly seen [124,125]. Through this technique, polydispersed samples can be analysed by easily tracking a range of different-sized particles simultaneously.

Inductively coupled plasma optical emission spectrometry (ICP-OES) allows valuable quantification of the concentration of dissolved ionic species in solution. This is highly useful, as the comparative dissolution of pristine particles against aged particles can act as a reciprocal marker for the remaining solid state of the affected ENM. However, the low expected concentration of ENMs and their dissolved constituents in environmental matrices will complicate measurements, and it is likely that some variables will fall below the lower limits of detection. ICP-MS (mass spectrometry) and single-particle ICP-MS have paved the way for particle number concentration to be determined directly [7]. This is a highly sensitive mass-based spectrometric method that requires minimal sample preparation, limiting the potential for human-induced transformation. Single-particle ICP-MS is also capable of high throughput and has been successfully used alongside field-flow fractionation (FFF) to measure the size and particle number concentrations of mixed Au (60 nm) and Ag (60 nm) along with bimetallic particles of Au and Ag composition combined [126]. As such, changes in particle size concentration will elucidate physical changes and behavioural patterns for specific ENMs. Like other analytical techniques, there are still limitations. For example, there is the persistent problem of lower limits of detection, though single particle ICP-MS does provide lower detection limits compared to other techniques [126]. If a significant particle size concentration of a sample falls below the limits of detection for the instrument, then this could again result in the misclassification of the ENM [7]. However, a study by Cascio et al. [127] found that detectable levels of Ag ENMs in consumer products using ICP-MS were found to be in close agreement with other instruments, including UV-Vis and TEM. This highlights the importance of carrying out multimethod characterisation in order to corroborate findings.

## 4. Knowledge Gaps and Future Challenges

Several knowledge gaps and future challenges exist around environmental transformations that must be overcome to ensure ENMs are produced using a safe-by-design approach. Despite recent advances in knowledge, the chemical, physical, and biological processes that drive transformations are still not fully understood. In addition, experimental studies are typically carried out with high ENM concentrations, which are not truly representative of environmental quantities and may elucidate mechanistic processes. These studies offer little value for predictive modelling in environmental settings with low-expected ENM concentrations. The toxicity behaviour of ENMs in natural aquatic systems differs from that found in spiked laboratory systems, with the concentration levels of ENMs in the latter usually being much higher. For instance, Lee et al. noted low acute toxicity of AgNPs and ZnONPs in zebrafish embryos in natural aquatic systems, while high acute toxicity was reported for spiked concentrations of AgNPs and ZnONPs [128]. Furthermore, it is not always possible to compare natural aquatic systems due to variations in biotic and abiotic factors as well as ecological and geological conditions.

Another inherent limitation is the current characterisation and detection methods employed for environmental transformation. Furthermore, ENMs are likely to undergo multiple transformations simultaneously, including oxidation/reduction and dissolution; interaction with NOM; and aggregation; as well as differential rates of degradation of surface coatings, making transformation a non-linear process [2,5]. For this reason, it is recommended that multiple solid-state characterisation methods be used to capture various processes at once and allow for an understanding of their effects on the properties of ENMs.

Access to data is profoundly limited, especially once ENMs have entered the consumer realm, where knowledge of specific uses, ageing processes, and disposal becomes much more scarce [10]. The variable nature of this data gives rise to various outcomes, making accurate modelling and future predictions of ENM behaviour extremely difficult to manage. Various mathematical models have been developed including material flow analysis and fate and transport models [11]. However, the models lack spatial resolution and require greater inputs of variables to account for the complex and dynamic nature of natural systems.

Despite progress in understanding ENM environmental transformations, many uncertainties remain, which pose challenges to current regulatory regimes. This is particularly relevant when ENMs undergo environmental transformations that may alter them from their pristine (as classified) state, especially where the rates of such processes may not be possible to constrain under current knowledge limitations.

## 5. Conclusions

The increased global production and use of ENMs is leading to aquatic and terrestrial contamination. Once released into the environment, ENMs undergo complex biotic and abiotic interactions, resulting in transformations that determine their fate, exposure concentration, form, behaviour, and toxicity. Transformations are changes that occur to the nanomaterial or its coating, the conformation of several nanomaterials, or all of these. To investigate the behaviour, fate, and toxicity of ENMs, the majority of studies have made use of spiked media with ENM concentrations higher than those usually representative of natural aquatic environments. However, such concentrations still demonstrate the potential threat of ENMs, which is crucial in regulating and managing their use. Initiatives, such as the Green Circular Economy, can help mitigate ENM levels in the environment. As a result, it would be beneficial if methods and technologies to recover ENMs and reuse them were developed.

This review focused on the issue of environmental transformations in ENMs. Consideration has been given to the likely chemical, physical, and biological transformations expected to occur to ENMs as they pass through different aquatic environments. The complexity of these transformations and the fact that they are so specific to a scenario due to their dependence on the properties of the ENM and environment were highlighted.

Chemical processes, including redox-driven dissolution, species-driven transformation, and photochemical alterations, have been considered for several relevant ENMs. Furthermore, processes of physical aggregation and biological transformation through biodegradation and biomodification have also been discussed for carbon-based nano-products and CeO_2_. Methods of detection and quantification have been highlighted, along with their benefits and limitations, such as the limits of detection and implications for regulatory classification.

Clearly, much work is required to progress our understanding of transformation processes in environmental media. Only through a complete understanding of ENM transformations throughout their entire lifecycle can we fully understand, model, and predict an ENM’s behaviour and potential toxicity. Once we are able to achieve this, we can proceed to inform safer design, production, and use of ENMs by advising the relevant scientific and regulatory stakeholders. 

## Figures and Tables

**Figure 1 nanomaterials-13-02098-f001:**
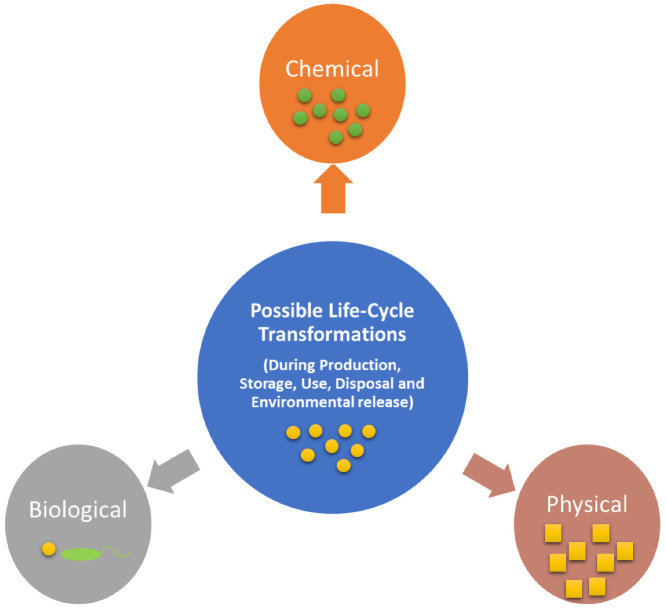
ENM transformations can result in chemical, physical, and biological changes.

**Figure 2 nanomaterials-13-02098-f002:**
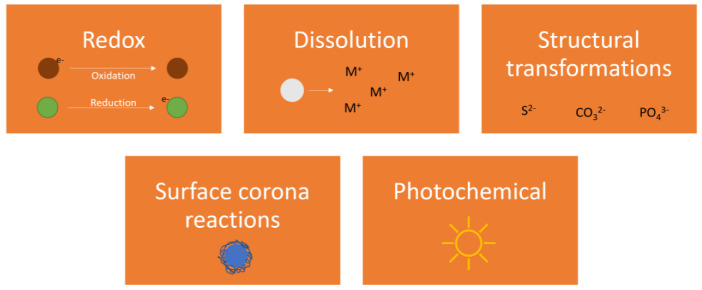
Dominant chemical transformations of ENMs in aquatic media.

**Figure 3 nanomaterials-13-02098-f003:**
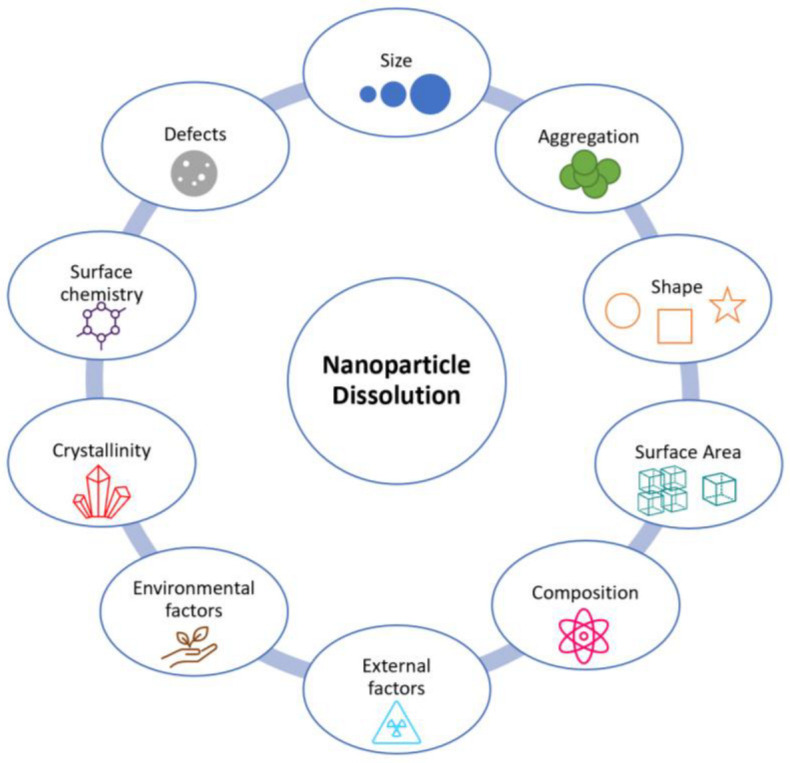
Properties that have the potential to influence ENM redox facilitated dissolution.

**Figure 4 nanomaterials-13-02098-f004:**
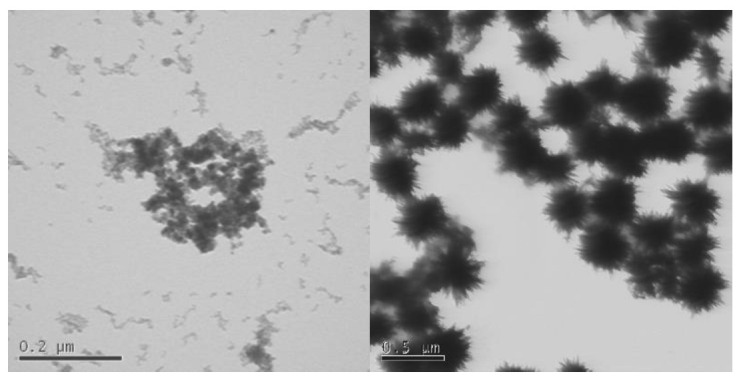
Ce_0.9_Zr_0.1_O_2_ before (**left**) (scalebar 0.2 μm) and after (**right**) (scalebar 0.5 μm) exposure to 5 mM phosphate solution (pH 5.5) for 3 weeks (unpublished image).

**Figure 5 nanomaterials-13-02098-f005:**
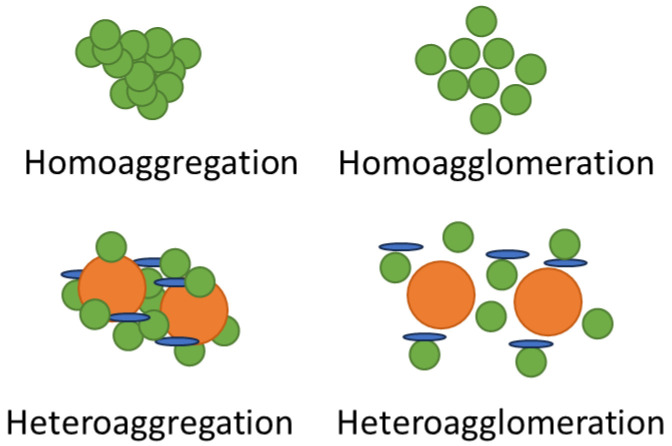
Comparison between aggregated and agglomerated particles, showing aggregates consisting of more tightly bound particles than agglomerates between the same type of nanomaterial (homo-) and different nanomaterials (hetero-).

**Figure 6 nanomaterials-13-02098-f006:**
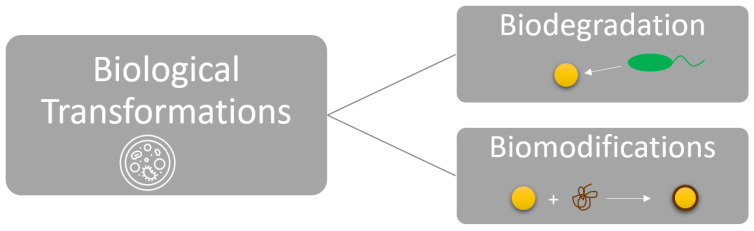
Categories of Biological Transformations for ENMs.

**Table 1 nanomaterials-13-02098-t001:** Factors determining the outcome of transformations and the specific property driving the transformation.

Factors Determining the Outcome of Transformations	Influencing Properties or Variables
**ENM morphology**	Size, shape, available surface area
**ENM chemistry**	Reactivity potential, possible reactions (e.g., oxidation), surface charge, aggregation state
**Environment**	pH, temperature, organic material

**Table 2 nanomaterials-13-02098-t002:** Examples of published transformation studies for commonly used ENMs in aquatic environments.

ENMs	Common Aquatic Environment Transformations	Examples from Literature
**TiO_2_**	Interaction with organic species and biomodification Photochemical reaction Redox Aggregation	[14,15,16,17,18] [19,20,21] [19,22] [15,16,18,21,23,24]
**ZnO**	Sulfidation Phosphitization Interaction with organic species and biomodification Photochemical Aggregation Dissolution	[25,26,27,28,29] [27,28,30,31] [14,32,33,34,35] [10,34] [32,33,36,37] [29,30,31,36]
**CuO**	Redox Dissolution Sulfidation Interaction with organic species and biomodification Aggregation	[38,39] [40,41,42,43,44,45] [38,40] [39,42,45,46,47,48] [41,42,45,48]
**CeO_2_**	Redox Phosphitization Interaction with organic species and biomodification Aggregation	[49,50,51] [51,52,53,54,55] [17,56,57,58] [51,57]
**Ag**	Redox Dissolution Sulfidation Carbonation Interaction with organic species and biomodification Aggregation Photochemical	[59,60,61,62] [59,61,63,64,65,66,67,68,69] [25,68,70,71,72,73,74,75,76] [60,77,78] [60,62,63,70] [63,78,79,80] [66,81]
**Graphene**	Photochemical reactions Aggregation Biodegradation Interaction with organic species and biomodification	[82,83,84,85] [83,85] [83,86] [82,83]

**Table 3 nanomaterials-13-02098-t003:** Specific examples of published transformation studies discussed in this review.

** Redox **		
**Nanomaterial**	**Observations**	**References**
**Cerium dioxide**	Valence state changes upon exposure to elevated temperatures	[2]
	Redox changes are dependent on exposure medium	[87,88]
**Elemental silver (Ag^0^)**	Undergo dissolution, facilitating the release of toxic Ag⁺ ions Significantly enhanced as pH is reduced	[10]
	An increase in system variables’ concentration, such as chloride and sulfide, will have a proportional effect on the rate of transformation Organic ligands, including NOM, may slow down redox processes.	[60]
**Zinc oxide**	Redox changes are dependent on exposure medium	[87,88]
**Cadmium sulfide**	Redox changes influenced by the presence of macromolecules and organic ligands from natural organic matter	[87,89]
** Dissolution **		
**Nanomaterial**	**Observations**	**References**
**Citrate-stabilised silver**	Dissolution influenced the toxicity	[90]
**Silver**	Dissolution rate increased at high ionic strength and low pH	[44]
**Copper oxide**	Dissolution is affected by the water characteristics—more soluble in deionised than natural pond water; however, the dissolution rate was faster in pond water compared to deionised water	[43]
**Zinc oxide**	Under oxic conditions, ZnO NPs were dissolved within a few hours. By contrast, ZnO NP dissolution under anoxic conditions was much slower.	[29]
** Sulfidation **		
**Nanomaterial**	**Observations**	**References**
**Silver**	Partly sulfidised Ag ENMs released fewer toxic Ag⁺ ions than pristine Ag ENMs over 48- and 120-h intervals.	[71]
	Increasing the presence of NOM suppressed the sulfidation of Ag nanowires in the aquatic environment.	[91]
	The presence of divalent cations compared to monovalent ions in solution, accelerated sulfidation rates	[91]
	Smaller AgNPs could result in an enhanced sulfidation rate owing to the reaction rate’s dependency on the specific surface area of the NP	[75]
	The increased HS^−^/Ag ratio and NOM presence influenced sulfidation. The presence of NOM was also found to influence the sulfidation of AgNPs.	[68,75]
	The presence of HA promoted sulfidation by replacing the surface coating, thus increasing the available surface area	[68]
	Sulfidation mitigated the toxicity of constructed wetlands	[76]
**PVP-coated Ag**	Sulfidation was found to reduce dissolution and limit toxicity	[72]
**Zinc oxide**	Stabilisation through sulfidation can reduce toxicity as it reduces dissolution and ion release	[26,28]
** Posphatization **		
**Nanomaterial**	**Observations**	**References**
**Zinc oxide**	pH-dependent and more likely in acidic environments than alkaline environments	[92]
	A decrease in toxicity in embryonic zebrafish	[28]
	Altered morphology	[31]
	Transformation products are larger than their pristine counterparts, and thus, surface reactivity is decreased, leading to reduced dissolution and muted toxic potential	[31]
**Cerium dioxide**	Physical and chemical changes occur	[53,54,55]
	Increased concentrations of phosphate will encourage desorption, limiting persistence and reducing toxicity risk	[10]
	Phosphate was capable of immobilising CeO_2_ through phosphate complexation in plant roots	[93]
** Carbonation **		
**Nanomaterial**	**Observations**	**References**
**Silver**	Inorganic silver carbonate (Ag_2_CO_3_) coatings have been applied as capping agents to stabilise them against aggregation	[60]
	At an alkaline pH, negatively charged CO_3_^2−^ surface capping could inhibit aggregation	[78]
** Surface Corona Reactions **		
**Nanomaterial**	**Observations**	**References**
**Silver**	Positively charged proteins enhanced the dissolution and sulfidation of AgNPs	[68]
**ZnO, TiO_2_, SiO_2_ and Al_2_O_3_**	Adsorption of HA was dependent on pH and decreased as the solution became more basic	[14]
**Titanium dioxide**	NPs with clay were toxic to zebrafish embryo development while NPs in the presence of HA displayed a protective effect	[18]
**Gold**	In low ionic-strength solutions, HA provided an additional coating, thereby providing additional resistance from pH induced aggregation	[94]
**Zinc oxide**	Reduction in toxicity in the presence of HA	[35]
** Photochemical Transformation **		
**Nanomaterial**	**Observations**	**References**
**Titanium dioxide**	UV irradiation in the environment significantly increased aggregation	[21]
**Graphene oxide**	Simulated sunlight can rapidly reduce the GO, producing by-products of CO_2_ and low-molecular weight species	[84]
**Silver**	NOM-facilitated photo-reduction of ionic Ag in river water, could precipitate NPs of different sizes and morphologies	[84]
**C-60**	Particles underwent surface oxidation and hydroxylation in the presence of dissolved O_2_	[95]
** Physical Transformations **		
**Nanomaterial**	**Observations**	**References**
**Surface coatings**	Surface coatings can be lost through biodegradation, which ultimately results in aggregation	[96]
	Using temperature as a proxy for ageing led to enhanced degradation of the PVP coating	[2]
**Silver**	Lower aggregation and higher particle stability was reported with increasing pH	[80]
**Copper oxide**	The aggregation and sedimentation of CuO NPs in soil solutions was influenced by the NP size and the soil properties	[48]
** Biodegradation **		
**Nanomaterial**	**Observations**	**References**
**C-60 (fullerenes)**	Not susceptible to biodegradation due to cage-structure	[10]
**Single walled carbon nanotubes**	Biodegradation observed when incubated with horseradish peroxidase and H_2_O_2_ via enzyme catalysis	[97]
** Biomodification **		
**Nanomaterial**	**Observations**	**References**
**Cerium dioxide**	Shape changes and presence of aggregation	[93]
**CNT**	Degradation of the lipid-coating which enabled the CNT to aggregate, aiding in their destabilisation	[98]
**Polystyrene**	NPs quickly acquired specific macromolecular coronas on their surfaces, which induced aggregation and increased uptake and gut retention	[99]
**Titanium dioxide**	NPs with EPS coronas adsorbed more heavy metals compared to NPs without EPS coronas	[17]
**Cerium dioxide**	NPs with EPS coronas adsorbed more heavy metals compared to NPs without EPS coronas	[17]

**Table 4 nanomaterials-13-02098-t004:** Summary table of the main characteristics of the transformations reviewed in this paper.

Transformation Classification	Type of Transformation	Key Features	Environmental Impact
**Chemical**	**Redox**	Influenced by media composition and conditions Dependent on intrinsic physicochemical properties of ENMs Influenced by the presence of macromolecules and organic ligands	Drives reactivity Internalisation may cause ROS production
**Chemical**	**Dissolution**	Used as a measure of bio-durability and toxicity	Enhanced by environmental parameters Enhances toxicity
**Chemical**	**Sulfidation**	Influenced by media composition and conditions Usually accompanied by substantial aggregation and sedimentation and a lower dissolution rate	Sulfidation can reduce toxicity in low redox environments, as it reduces the dissolution
**Chemical**	**Phosphatization**	pH dependent and more likely in acidic environments than alkaline environments Decreases particle surface reactivity	Drives particle size increase and reduces dissolution. Can mitigate toxicity
**Chemical**	**Carbonation**	Affects particle stability	Inhibits aggregation
**Chemical**	**Surface corona reactions**	ENM surface properties modified Positively charged proteins enhance dissolution and sulfidation	Biomolecules may result in increased biocompatibility of ENMs, thereby also altering toxicity
**Chemical**	**Photochemical**	May increase aggregation	Generates ROS and oxidative tissue damage
**Physical**	**Aggregation and agglomeration**	Potential decrease in reactivity and toxicity, resulting from an increase in size	Increases environmental persistence
**Biological**	**Biodegradation**	May break down ENMs into less harmful counterparts	Decreases environmental persistence
**Biological**	**Biomodification**	May convert ENMs to more biocompatible forms	Potentially increases bioavailability and bioaccumulation; may also increase toxicity

**Table 5 nanomaterials-13-02098-t005:** ENM detection methods along with information regarding suitable ENM categories, the main information gained, and advantages and disadvantages.

Characterisation Method	ENM Types Suitable for the Method	Main Information Gained	Advantages	Disadvantages
**Electron microscopy (SEM, TEM, STEM)**	Inorganic ENMs	Morphological and compositional information (quantitative)	Allows direct comparisons between pristine and aged ENMs. Size, morphology, and surface conditions can be studied. Chemical composition provided by EDX. Structural information is available from TEM/STEM.	Ex-situ methods. iSEM may be unsuitable for ultrasmall ENMs. Laborious sample preparation. 2D representation of a 3D material. Operator bias.
**Dynamic Light Scattering (DLS)**	Can be applied to any ENM category but cannot differentiate between different ENMs if present as mixtures	Provides information on the hydrodynamic size (Hd) and zeta potential (ζ) and allows the determination of particle size distribution.	Gives information on hydrodynamic size (Hd) and zeta potential (ζ).	Cannot distinguish between isolated particles, clusters, and mixtures of sizes, or particles with different compositions. Assumes all particles are spherical.
**Nanoparticle Tracking Analysis (NTA)**	Suitable for visualising and analysing inorganic and organic particles in suspension; Cannot differentiate between different ENMs if present as mixtures	Can track different sized particles simultaneously.	Allows visual tracking of particles, allowing observation of aggregates. Can also track different-sized particles simultaneously.	Requires an optimum concentration for analysis. It cannot distinguish between individual particles and agglomerates/aggregates.
**Inductively coupled plasma mass spectrometry/optical emission spectrometry (ICP-MS, ICP-OES)**	Suitable for detecting and quantifying heavy (inorganic) elements.	Directly measure the particle number concentration (in single particle mode).	Allows quantification of dissolved ions in solution and directly measures the particle number concentration.	Signal intensity varies with each isotope. Interferences can occur when plasma-formed species have the same mass as the ionised analyte species. May require prior knowledge of particle composition
**Ultraviolet-visible spectroscopy (UV-Vis)**	Can be used to determine concentrations and provide information about the physical and electronic structures of both organic and inorganic compounds.	Provides information on the optical properties, size, concentration, and agglomeration state of the nanoparticles.	Can provide quantitative and qualitative analysis. Easy to handle and use. Non-destructive.	Unable to analyse compounds that do not interact with light in the UV and visible areas of the spectrum. Can only be used to characterise suspensions and is not suitable to measure solid or gaseous samples.
**Attenuated Total Reflection—Fourier Transform Infrared Spectroscopy (ATR-FTIR)**	Can provide information for both inorganic and organic substances.	Provides information on the surface composition and ligand binding.	Capable of analysing nanomaterial samples in suspension or powder form. Does not require ultra-high vacuum conditions.	Limited spatial resolution.
**X-ray Photoelectron Spectroscopy (XPS)**	Inorganic coatings, but can also be extended to natural organic coatings.	Determination of chemical composition surfaces and oxidation states.	Allows for the determination of electronic structure, elemental composition, and oxidation states. Highly sensitive to surface modifications.	Requires ultra-high vacuum conditions. Overlapping peaks in the spectra can complicate surface analysis. Contamination by adsorbed water or volatile organic compounds makes carbon and oxygen analysis difficult.
**High-Performance Liquid Chromatography (HPLC)**	Quantifies ligands conjugated to soft nanoparticles such as polymers and liposomes.	Suitable for the separation of nanoparticles in mixed samples or for the evaluation of the nanoparticle surface upon interaction with the stationary phase.	Can detect and measure very small amounts of a substance. Can separate and analyse a sample quickly.	Nanoparticles may stick to the column. Limited sample size, which can make it difficult to analyse large samples or many samples at once.
**Liquid Chromatography-Mass Spectrometry (LCMS)**	Used to characterise colloids in natural organic matter but can also be used to study trace metal distributions over colloidal particles.	Quantifies ligand density through cleavage of the surface ligands from the nanoparticles.	Robust analytical technique that provides the high sensitivity and selectivity required to detect the exact molecular weight of a wide range of samples. Can separate and identify solutes in low concentrations (in parts per million) in a complex mixture. Major advantage of LCMS over HPLC is that LCMS can achieve a complete elucidation of the chemical structure of the molecule.	Only works with volatile buffers. Residual impurities being analysed should be ionised.

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
