# Peer review of "A Review of the Aquatic Environmental Transformations of Engineered Nanomaterials"

_nanomaterials, 2023, doi:10.3390/nano13142098_

Round 1
Reviewer 1 Report
This paper reviews the aquatic environmental transformations of engineered nanomaterials (ENMs), and it outlines the key influences and outcomes of ENMs evolution pathways in aquatic environments and provides an assessment of potential environmental transformations, focusing on key chemically, physically, and biologically mediated processes. I have following comments to make for the benefit of the authors of this manuscript.
1. The English of the manuscript should be carefully polished, and some expression should be improved.
2. The writing of this paper is rough, and some descriptions are difficult to understand.
3. The comparison of different articles is not well presented, so no in-depth comparison in a review paper is not acceptable.
4. A good review paper needs to cover the current state, and it should also include some insight given into the future of the topic on which the review is based.
5. For a review paper, the authors need to bring some criticisms, own opinions and new research directions. I encourage the authors to add as many criticisms, own opinions and new research directions as possible.
Author Response
The authors thank the constructive comments made by the reviewer. On taking on the suggestions the authors have been able to strengthen their review and the overall idea and message.
This paper reviews the aquatic environmental transformations of engineered nanomaterials (ENMs), and it outlines the key influences and outcomes of ENMs evolution pathways in aquatic environments and provides an assessment of potential environmental transformations, focusing on key chemically, physically, and biologically mediated processes. I have following comments to make for the benefit of the authors of this manuscript.
- The English of the manuscript should be carefully polished, and some expression should be improved.
The manuscript has been careful proof-read. All the language changes have been tracked throughout the manuscript.
- The writing of this paper is rough, and some descriptions are difficult to understand.
The manuscript has been reviewed and proof-read. In addition to changes to the language, changes have also been made to some descriptions in order to make them easier to understand.
- The comparison of different articles is not well presented, so no in-depth comparison in a review paper is not acceptable.
The comparisons have now also been presented in Table format and the paper has been revised to further the comparisons.
- A good review paper needs to cover the current state, and it should also include some insight given into the future of the topic on which the review is based.
The authors have reviewed the paper and highlighted further the need for an understanding of environmental transformations through the set-up of standard guidelines.
- For a review paper, the authors need to bring some criticisms, own opinions and new research directions. I encourage the authors to add as many criticisms, own opinions and new research directions as possible.
The authors thank the reviewer for this comment. The manuscript has been revised to include criticisms and opinions highlighting the complexity and importance of aquatic transformations. Furthermore, the authors have shared their thoughts on how the field should progress as well as considerations that need to be made.

Reviewer 2 Report
My decision: Major Revision
The author summarizes numerous relevant literatures and provides an overview of the main impacts and outcomes of the ENMs evolutionary pathway in aquatic environments, as well as an assessment of potential environmental transitions. The author mainly elucidates key chemical, physical, and biological mediated processes. The author has done a lot of related literatures collection and analysis work. However, many problems still need to be solved. I think it can be accepted and published after solving some questions below:
1. The author mentioned “Pourbaix diagrams” in the article, please indicate the source of the figure and refer to the references. The relevant details of the figure were not found in the article.
2. In the introduction section, the author mentioned the distribution of ENMs in various media. Please unify the number of decimal places after the decimal point in the percentage.
3. The author mentioned two spectral analysis methods, ICP-OES and ICP-MS, in section 2.2.2. It is recommended to provide a brief introduction to the above methods (including working principles and applicability), as well as how to determine the content of dissolved ionic substances, for the convenience of future readers' learning.
4. The author has repeatedly cited literature to confirm that the presence of NOM can affect or inhibit the sulfurization of Ag (ENM). It is recommended to clarify the reason for this result (explain the inhibition principle), which is convenient for subsequent readers to understand, deepen the readability of the article, and avoid confusion for readers.
5. Please increase the resolution of Figure 4. The current version is not clear and difficult to read.
6. The author mentioned in the article: “…. the growth of hydroxyl surface groups as identified with attenuated total reflectance-Fourier transform infrared spectroscopy (ATR-FTIR).” Can you specify how the two are related?
7. In Part 3, can the author associate the detection method with specific ENM categories, and which ENM categories are suitable for which method?
Author Response
The authors thank the reviewer for their comments and have tackled the suggested changes and improvements.
The author summarizes numerous relevant literatures and provides an overview of the main impacts and outcomes of the ENMs evolutionary pathway in aquatic environments, as well as an assessment of potential environmental transitions. The author mainly elucidates key chemical, physical, and biological mediated processes. The author has done a lot of related literatures collection and analysis work. However, many problems still need to be solved. I think it can be accepted and published after solving some questions below:
- The author mentioned “Pourbaix diagrams” in the article, please indicate the source of the figure and refer to the references. The relevant details of the figure were not found in the article.
On reviewing the paper, the authors reworded that paragraph to make it clearer as part of the response to a comment made by reviewer 1. The authors removed any reference to Pourbaix diagrams.
- In the introduction section, the author mentioned the distribution of ENMs in various media. Please unify the number of decimal places after the decimal point in the percentage.
The number of decimal places has been unified such that the percentages mentioned between lines 51-53 all have one decimal point.
- The author mentioned two spectral analysis methods, ICP-OES and ICP-MS, in section 2.2.2. It is recommended to provide a brief introduction to the above methods (including working principles and applicability), as well as how to determine the content of dissolved ionic substances, for the convenience of future readers' learning.
A brief description of ICP-OES and ICP-MS, including working principles, applicability as well as how to determine the content of dissolved ionic substances, has been included in the manuscript in Section 2.2.2. A new reference (reference 95) has also been added.
- The author has repeatedly cited literature to confirm that the presence of NOM can affect or inhibit the sulfurization of Ag (ENM). It is recommended to clarify the reason for this result (explain the inhibition principle), which is convenient for subsequent readers to understand, deepen the readability of the article, and avoid confusion for readers.
The reason for the inhibition (or promotion) of sulfurization of Ag (ENM) by NOM has been better clarified in the new Section 2.2.3.1 (Sulfidation).
- Please increase the resolution of Figure 4. The current version is not clear and difficult to read.
Figure 4 has been replaced with a new image such that the resolution could be improved.
- The author mentioned in the article: “…. the growth of hydroxyl surface groups as identified with attenuated total reflectance-Fourier transform infrared spectroscopy (ATR-FTIR).” Can you specify how the two are related?
ATR-FTIR provides information related to the presence of specific functional groups, as well as the chemical structure of polymer materials. Shifts in the frequency of absorption bands and changes in relative band intensities indicate changes in the chemical structure or changes in the environment around the sample. Thus, ATR-FTIR spectroscopy can be used to determine the resultant surface chemistry especially following induced chemical or physical modifications (Anderson, J. M., and G. Voskerician. "The challenge of biocompatibility evaluation of biocomposites." Biomedical Composites. Woodhead Publishing, pp. 325-353, 2010).
This has been better re-worded and made clearer in Section 2.2.5 of the manuscript. A new reference (reference 110) has also been included to help with the explanation.
- In Part 3, can the author associate the detection method with specific ENM categories, and which ENM categories are suitable for which method?
A new table has been included in Section 3 of the manuscript listing the different characterisation techniques mentioned in this manuscript together with the main information gathered from each technique and which ENM categories would be suitable for such techniques.
Reviewer 3 Report
This review paper describes the aquatic environmental transformation of engineered nanomaterials. In this review paper, the authors discussed various ways of transformation including, chemical, physical, and biological transformations. The authors also explained the various methods of chemical transformations including, redox, dissolution, structural, surface corona reactions, and photochemical reactions.
The paper requires revision for the following.
1. The authors should add details about the engineered nanomaterials and how they differ in characteristics in the form of a table.
2. The authors should add the pros and cons of various transformation approaches methods in the form of a table.
3. The authors should emphasize in the abstract and conclusion why this study is important and how it will contribute to the field.
4. Figure 1 should be revised for more clear imaging and bigger font.
5. The numbering of the sections should be revised as
2.2.1. Redox
2.2.2. Dissolution
2.2.3. Structural transformation
2.2.3.1. or i Sulfidation (it is a subsection of structural transformation)
2.2.3.2. or ii Phosphatization (it is a subsection of structural transformation)
2.2.3.3. or iii Carbonation (it is a subsection of structural transformation)
2.2.4. Surface Corona Reaction
2.2.5. Photochemical
6. For all types of transformations, the authors should add tables citing the papers already published describing the particular transformation and its summarized results.
7. All the old references should be replaced with recent ones, especially those published within the last five years.

Author Response
The paper requires revision for the following.
- The authors should add details about the engineered nanomaterials and how they differ in characteristics in the form of a table.
Figure 3 highlights the properties on ENMs with specific reference to dissolution. Over and above this Table 1 has been added to highlight the specific property of ENMs influenced by factors influencing transformations. Details about the ENMs and how they differ in characteristics cannot be added as a table as this varies from one example to another, and each example should be considered individually.
- The authors should add the pros and cons of various transformation approaches methods in the form of a table.
A new Table has been added to address this.
- The authors should emphasize in the abstract and conclusion why this study is important and how it will contribute to the field.
The following sentences have been added to the abstract to emphasize the contribution: Through obtaining a full understanding of the potential environmental transformations that ENMs can undergo we can model and predict their behaviour. This will, in turn, shed light on their potential impact and help to inform regulatory bodies as deemed suitable.
The final sentence of the conclusion has been updated to: Once we are able to achieve this, we can proceed to inform safer design, production and use of ENMs through advising the relevant scientific and regulatory stakeholders.
- Figure 1 should be revised for more clear imaging and bigger font.
The Figure has been revised accordingly.
- The numbering of the sections should be revised as
2.2.1. Redox
2.2.2. Dissolution
2.2.3. Structural transformation
2.2.3.1. or i Sulfidation (it is a subsection of structural transformation)
2.2.3.2. or ii Phosphatization (it is a subsection of structural transformation)
2.2.3.3. or iii Carbonation (it is a subsection of structural transformation)
2.2.4. Surface Corona Reaction
2.2.5. Photochemical
The numbering has been edited as suggested.
- For all types of transformations, the authors should add tables citing the papers already published describing the particular transformation and its summarized results.
The authors have added Table 3 in order to tackle this suggestion.
- All the old references should be replaced with recent ones, especially those published within the last five years.
A thorough literature search has been carried out in order to ensure that all the latest references have been included. In some cases older references needed to be included due to the lack of more recent references discussing specific examples of transformations.
Round 2
Reviewer 1 Report
The manuscript's quality has been improved. I recommend its acceptance for publication in its present form.
Reviewer 2 Report
A review of the aquatic environmental transformations of engineered nanomaterials-R1
My decision: Accept
The author summarizes numerous relevant literatures and provides an overview of the main impacts and outcomes of the ENMs evolutionary pathway in aquatic environments, as well as an assessment of potential environmental transitions. The author mainly elucidates key chemical, physical, and biological mediated processes. The author revised and improved the article carefully according to the requirements of the reviewers, and did a lot of work. The readability of the article has been significantly improved. I am satisfied with the revised version and the author’s reply. I think this version can be accepted and published.